# Interactome Mapping of eIF3A in a Colon Cancer and an Immortalized Embryonic Cell Line Using Proximity-Dependent Biotin Identification

**DOI:** 10.3390/cancers13061293

**Published:** 2021-03-14

**Authors:** Diep-Khanh Vo, Alexander Engler, Darko Stoimenovski, Roland Hartig, Thilo Kaehne, Thomas Kalinski, Michael Naumann, Johannes Haybaeck, Norbert Nass

**Affiliations:** 1Department of Pathology, Medical Faculty, Otto-von-Guericke University Magdeburg, D-39120 Magdeburg, Germany; diepkhanh.vo@gmail.com (D.-K.V.); Darko.Stoimenovski@st.ovgu.de (D.S.); Thomas.Kalinski@med.ovgu.de (T.K.); Johannes.Haybaeck@i-med.ac.at (J.H.); 2Institute of Experimental Internal Medicine, Medical Faculty, Otto von Guericke University, D-39120 Magdeburg, Germany; Alexander.engler@med.ovgu.de (A.E.); Thilo.Kaehne@med.ovgu.de (T.K.); naumann@med.ovgu.de (M.N.); 3Institute of Molecular and Clinical Immunology, Otto von Guericke University Magdeburg, Leipziger Str. 44, D-39120 Magdeburg, Germany; roland.hartig@med.ovgu.de; 4Department of Pathology, Neuropathology, and Molecular Pathology, Medical University of Innsbruck, A-6020 Innsbruck, Austria; 5Department of Pathology, Diagnostic & Research Center for Molecular BioMedicine, Institute of Pathology, Medical University of Graz, A-8010 Graz, Austria; 6Center for Biomarker Research in Medicine, A-8010 Graz, Austria

**Keywords:** eukaryotic translation initiation factor, interactome, BioID, cancer, ER-stress

## Abstract

**Simple Summary:**

The behavior of a cancer cell is greatly influenced by its proteome, which is the result of protein biosynthesis, modification and degradation. Eukaryotic initiation factors control protein biosynthesis and exhibit complex interactions both with each other and with the protein-coding mRNAs. Thus, the detailed molecular interactions of the eIFs might be crucial for the aggressiveness of a cancer. Here, we applied the BioID method, to analyze the interactions of eIFs using eIF3A as a prototype. As a result, we identified several known interactors of eIF3A and detected proteins not described in this context before. In particular, the occurrence of proteins involved in proper folding suggests a close coupling of protein biosynthesis and folding in cancer. This method is therefore a promising tool for further analysis of the protein biosynthesis.

**Abstract:**

Translation initiation comprises complex interactions of eukaryotic initiation factor (eIF) subunits and the structural elements of the mRNAs. Translation initiation is a key process for building the cell’s proteome. It not only determines the total amount of protein synthesized but also controls the translation efficiency for individual transcripts, which is important for cancer or ageing. Thus, understanding protein interactions during translation initiation is one key that contributes to understanding how the eIF subunit composition influences translation or other pathways not yet attributed to eIFs. We applied the BioID technique to two rapidly dividing cell lines (the immortalized embryonic cell line HEK-293T and the colon carcinoma cell line HCT-166) in order to identify interacting proteins of eIF3A, a core subunit of the eukaryotic initiation factor 3 complex. We identified a total of 84 interacting proteins, with very few proteins being specific to one cell line. When protein biosynthesis was blocked by thapsigargin-induced endoplasmic reticulum (ER) stress, the interacting proteins were considerably smaller in number. In terms of gene ontology, although eIF3A interactors are mainly part of the translation machinery, protein folding and RNA binding were also found. Cells suffering from ER-stress show a few remaining interactors which are mainly ribosomal proteins or involved in RNA-binding.

## 1. Introduction

Protein biosynthesis, a highly organized and tightly regulated process, undergoes three phases: initiation, elongation, and termination. It is currently accepted that protein biosynthesis is mainly regulated in the initiation phase that organizes the assembly of the translationally active 80S ribosome [1]. Notably, this control is not only quantitative, but also qualitative, as structural elements of the mRNA are important for translation efficiency. One well-known example is the switch between m^7^G-cap-dependent and the IRES-mediated initiation of translation [2]. It facilitates translation under cellular stress [3] and is also widely used for translation of viral proteins [4]. The initiation phase is controlled by an interaction between the eukaryotic initiation factors and the structural elements of the mRNA. About 33 eIF proteins and 5 auxiliary factors are known and combined in complexes that all have different functions in the assembly of the ribosome and in the scanning of the mRNA for the translational start point [5]. eIF3 is the largest complex, consisting of 11–13 subunits forming an 800 kDa complex. eIF3A, together with eIF3C, -E, -I, -K, -L, -F and –M, forms the core subunit of eIF3 [6,7]. This complex is involved in binding the 40S ribosome, eIF1, eIF4G and eIF5, and plays a role in attaching the 43S ribosomal complex to the mRNA for subsequent scanning for the translational start codon. This process is accompanied by a rearrangement of peripheral eIF3 subunits [8]. The topology of the 48S preinitiation complex, including eIF3A, was established in detail by cryo-electron microscopy [9], and the eIF3A interactome was characterized by immune-precipitation followed by mass spectrometric identification of attached proteins in fission yeast [10]. eIF3 plays a role in transcript-specific translation regulation [11], and eIF3 subunits bind directly to the HCV IRES element [12,13,14] and other secondary structures of mRNAs [15]. It is also involved in the repression of ferritin light chain translation [16]. Another binding partner of eIF3 is the RNA polymerase subunit hRPB11 [17].

Protein biosynthesis is frequently dysregulated not only in cancer [18,19,20], but also in ageing [21,22], which implies both quantitative and qualitative aspects. Interestingly, the composition of eIFs can be different in cancer entities [23,24,25], and this seems to be the basis for a differential control of protein biosynthesis and, as a consequence, for the proteome of cancer cells [26]. A quantitatively different abundance of the eIF subunits might result in alterations in the composition of initiation complexes and in increased amounts of unbound eIFs that might have functions other than ribosome assembly. Notably, an unexpected function in RNA splicing has recently been demonstrated for eIF2α [27,28].

Protein biosynthesis is also regulated by several signal transduction pathways [23]. Firstly, eIF2α is the target of four kinases (eIF2α kinases 1–4) that are regulated for example by ER-stress [29]. Inhibition of dephosphorylation by compounds, such as salubrinal, increases eIF2α phosphorylation, thus decreasing protein biosynthesis, which can be beneficial in the case of human diseases accompanied by ER-stress [30]. Furthermore, it has an impact on drug sensitivity [31]. Secondly, the TOR-pathway controls the activity of the eIF4 complex by phosphorylation of eIF4 binding proteins, thereby linking metabolism to translation [32]. The assembly of the eIF4 complex can be targeted by compounds, such as 4-EGI [33] or rocaglates [34,35]. Taken together, dysregulated protein biosynthesis represents a suitable target for the treatment of diseases, and a deeper understanding of the protein-RNA interactions during translation initiation might contribute to the development of novel targeted therapies, especially for cancer [36].

Here, we aimed to determine the eIF3A interactome by applying the BioID technique to two fast-growing cell lines, the human embryonic kidney cell line HEK-293T and the colon cancer cell line HCT-116, under optimal growth conditions and under severe stress caused by thapsigargin, an ER-stress-inducing compound. Performing these experiments, we intended to find previously unknown interacting proteins that might be mediators of translational regulation or indicate new functions of eIF3A.

## 2. Results

### 2.1. Identification of Cellular Proteins Biotinylated by BirA*-eIF3A

EIF3A, the largest subunit of eIF3, has been reported to be highly expressed in malignant tumors [37], and its abundance correlates with dedifferentiation, continuously increasing from normal mucosa via hyperplastic to neoplastic transformation in colorectal cancer [38]. It is known that in most cases, EIF3A interacts with other eIFs subunits during the initiation of translation. Here, we applied the BioID proximity-based biotin labeling technique [39,40] to identify interacting proteins of eIF3A in two human cell lines: the human colorectal carcinoma cell line HCT-116 and the human embryonic kidney cell line HEK-293T. These cell lines differed slightly in the ratio of m^7^G-cap- to Polio-IRES-mediated translation (Appendix A), with the cancer cell line HCT-116 showing a small but significantly higher IRES/cap ratio.

For BioID, we generated a construct with eIF3A fused to the C terminus of BirA*. Expression of this fusion protein shifted the IRES/cap ratio towards Polio-IRES-mediated translation, which was similar to the effect of eIF3A overexpression (Appendix A). We therefore think that this fusion protein functionally integrates itself into the protein translation machinery. The fusion construct was transfected into the cells, and biotinylation stimulated by adding external biotin, followed by a streptavidin pull-down of biotinylated proteins. Immunoblotting was performed to confirm the expression of the BirA*-eIF3A fusion construct. After analyzing the labeling kinetics, we decided to incubate both cell lines with biotin for 6 h and 24 h and, furthermore, to subject samples from both time points to further analysis. Use of the control vector BirA* resulted in a strong signal of auto-biotinylated BirA* with an apparent molecular mass of 37 kDa in both cell lines. We detected and enriched additional biotinylated proteins using the streptavidin pull down (Figure 1A). Transfection with the BirA*-eIF3A fusion construct resulted in the detection of an additional biotinylated protein at 210 kDa, representing the BirA*-eIF3A fusion protein. Using an anti-eIF3A antibody for these samples, a protein of the same size was detected in addition to the endogenous eIF3A protein (166 kDa), whereas this fusion protein was absent in control BirA* transfections. In streptavidin-purified protein samples, only the fusion protein was detected (Figure 1B). The eIF3A-BirA* fusion protein resulted in less biotinylated proteins compared to the BirA* controls.

We were able to further confirm the BioID-mediated biotinylation by immunofluorescence (Figure 2). Transfected, biotin-incubated cells showed a strong signal for biotinylated proteins where the BirA*-eIF3A protein exhibited a different staining pattern compared to the BirA* control transfections. BirA*-eIF3A produced a stronger signal mainly in the cytosol and the perinuclear region, whereas BirA* resulted in an even staining in cytosol and nucleus. We interpret this pattern as a localization of eIF3A to the rough ER as the major site of protein biosynthesis.

### 2.2. eIF3A Interacting Proteins

To identify eIF3A-proximal proteins, we performed mass spectrometry of purified biotinylated proteins for cells transfected with the BirA*-eIF3A vector and the BirA* control vector alone, both in HCT-116 and HEK-293T cells and in non-transfected cells. We compared BirA*-eIF3A to BirA* control, both stimulated with biotin for 6 h and 24 h. Investigating HCT-116 cells, we identified 1057 proteins subjected to biotinylation for 6 h and 1028 proteins subjected to 24 h of biotinylation. The investigation of HEK-293T cells revealed 1583 proteins with 6 h biotinylation and 1541 proteins with 24 h biotinylation. We then applied a filtering process where proteins were required to have more than three unique peptides identified in BirA*-eIF3A and at least one unique peptide in the BirA* control; proteins that had misconducting ratios between unique BirA*-eIF3A- and BirA* control peptides (BirA* > BirA*-eIF3A) were not considered for analysis. Additionally, we excluded keratins from the list, as these proteins are frequently a result of exogenous contamination. Finally, we also excluded naturally biotin-bearing proteins, such as carboxylases, as we observed an unspecific enrichment in previously performed BioID experiments.

Next, we applied the ProgenesisQI software to the data of the remaining proteins to identify significant changes between the cell lines and treatments. This software allows for a comparative abundance profiling from multiple h.p.l.c. MS runs for different experimental conditions in a label-free manner (LFQ). MS data of each cell line and experimental batch was processed to increase the accuracy of the calculations. Proteins were only considered if the mean ratio between BirA*-eIF3A and BirA*-control was higher than 1.5 (50% increase in abundance) in at least one experiment. After all these filtering steps, we compiled the final list of 84 proteins and compared the HCT-116 and HEK-293T cells for the overlaps of these proteins identified within 6 h or 24 h (Figure 3a). The HCT-116 cells revealed 24 candidate proteins in the 6-h biotinylation group, while the HEK-293T cells showed 51 candidates with 10 proteins being present in both cell lines. The HCT-116 cells having undergone a 24 h biotinylation exhibited 17 proteins, while 44 proteins were detected in HEK-293T cells, and 10 were common to both cell lines. Based on these data, we built a heatmap illustrating the ratio between BirA*-eIF3A and the BirA* control (Figure 3b) for these two cell lines. The highest score was obtained for eIF3A itself, which is consistent with self-biotinylation of the fusion protein. BirA*-eIF3A also gave high scores for other eIF3 subunits, mostly for eIF3C, -D, -F, -G, which proved the expected close spatial relationship of these proteins.

To further corroborate our results and to obtain insights into initiation complexes under the condition of a severe downregulation of protein biosynthesis, we applied thapsigargin, a potent ER-stress inducer [41], to both cell lines and determined the eIF3A-birA*-dependent biotinylated proteins. Thapsigargin resulted in increased PERK- and IRE1α-phosphorylation in both cell lines, but induction of CHOP (CCAAT-enhancer-binding protein homologous protein) became evident only in HCT-116 cells. There was also a slight increase in eIF2α-phosphorylation. (Appendix A). As seen in the heatmap (Figure 3b), the number of eIF3A proximal proteins has decreased dramatically. There were only 4 and 3 proteins that were significantly labeled after 6 h of biotinylation in HCT-116 and HEK-293T cells, respectively, and two and three proteins after 24 h. One of these proteins was eIF3A, with scores similar to those in the controls, which proves that biotinylation still occurs under ER-stress. Compared to controls, the other proteins exhibiting unaltered eIF3A-dependent biotinylation ratios were RBM39 and RS17 in HCT-116 cells and ATPO and HORN in HEK-293T cells after 6 h of biotinylation. After 24-h biotinylation, only RS17 remained constant in HCT-116 cells, whereas RS17 and RS29 remained unaltered in HEK-293T-cells. A few proteins appeared only under conditions of ER-stress and only in limited experimental conditions (DSG1, H4, DHX9 and DCD).

We performed a co-immunoprecipitation as well as immunofluorescence co-localization experiments to obtain further evidence for putative eIF3A interactions identified by BioID (Appendix A). For these experiments, we selected eIF3D, eIF4A and YB-1, as well as Cyclophilin A (PPIA). For eIF3D, eIF4A1 and YB-1, using the co-immunoprecipitation approach, we indeed confirmed a physical interaction. However, we failed to obtain a specific signal in the co-immunoprecipitation for PPIA. In immunofluorescence, eIF3A was detected in cytosol as well as the nucleus, as predicted. However, depending on the antibody used, the signal was more pronounced in the cytosol (rabbit monoclonal antibody) or the nucleus (mouse monoclonal antibody). We propose that the apparent different localization of eIF3A visualized using these two antibodies depends on the different epitopes that might be masked differently in the two compartments, maybe due to different binding partners. Additionally, the antibodies might react differently to the fixation conditions. Nevertheless, within the limited spatial resolution of this technique, eIF3A co-localized with eIF4A1, eIF3D, Cyclophilin and YB-1 either in nucleus or cytosol (Appendix A).

### 2.3. Gene Ontology Analysis of eIF3A-Proximal Proteins

The selected proteins (see Figure 3B) were then analyzed for overrepresentation in the gene ontology terms by using the Genecodis bioinformatics platform [42]. We considered those terms as significant that had at least three annotated proteins (Table 1) and an adjusted *p*-value < 0.01. Additionally, the eIF3A-dependent biotinylation ratios are presented as box plots (Figure 4 and Appendix A). A detailed list of the overrepresented GO-terms can be found in the Appendix A. The separate overrepresentation analysis data of the two cell lines is presented in the Appendix A.

For the GO-category “cellular component”, we found that most proteins were associated not only with cytosols and nuclei, but also with an over-representation in exosomes, focal adhesions or granules. Most of these significant overrepresentations disappeared under thapsigargin treatment (Figure 4 and Appendix A). For the GO category “molecular function”, over-representation mainly referred to protein biosynthesis and RNA binding, but cell adhesion and protein folding were also significant. Under Thapsigargin, RNA and nucleotide binding became the most prominent hits (Figure 4 and Appendix A). Within the term “biological process”, mainly protein biosynthesis, but also protein folding and gluconeogenesis, were overrepresented. Under Thapsigargin, significance values dropped, but besides translation, RNA binding and processing remained significant (Figure 4 and Appendix A).

### 2.4. Interaction Network of eIF3A

In addition to the GO term analysis, we used the cytoscape software [43] to visualize the network of proximal proteins and included known interactions from the string database. Most of the interacting proteins could indeed be integrated into a network. Nevertheless, there are some outliers that have no, or only a few, connections to the other proteins in the network. This holds especially true for DCD, YBX3 and MKL2 (Figure 5).

**Figure 5 cancers-13-01293-f005:**
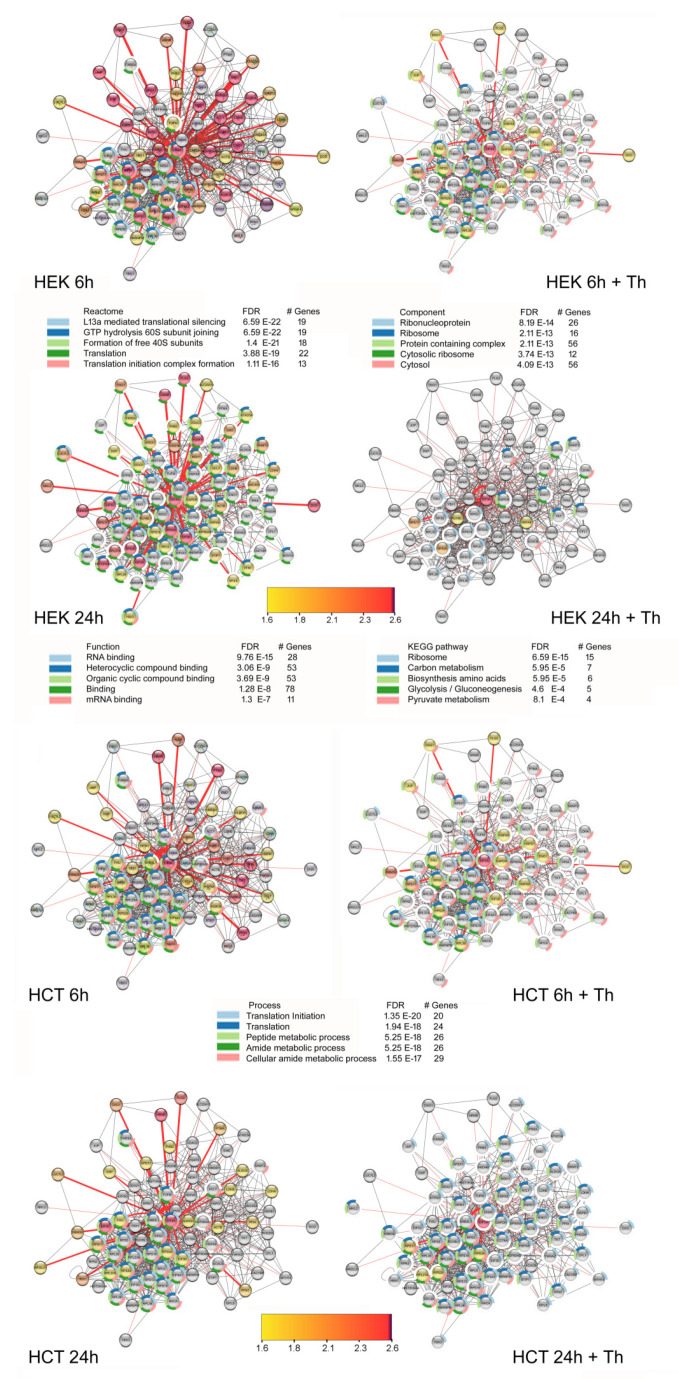
Visualization of the eIF3A-dependent biotinylation ratios by cytoscape [43] for the cell lines HEK-293T (HEK) and HCT-116 (HCT) under all experimental conditions incorporating interactions from the string database [44]. For this visualization, all proteins were included exhibiting a biotinylation ratio higher than 1.5 in at least one condition. The color of the protein nodes corresponds to the biotinylation ratios according to the color gradient from yellow to purple as shown in the figure. For grey nodes, the ratio is smaller than 1.5. Red lines indicate interactions determined in this study; black lines indicate interactions taken from the string database. The width of the red lines corresponds to the biotinylation ratio. FDR: false discovery rate. The result of the GO-term enrichment analysis for the cellular component, molecular function and biological process and for the databases KEGG pathway [45] and Reactome [46] is also presented.

## 3. Discussion

eIFs are frequently dysregulated in cancer cells, and this might provide the basis for quantitative and qualitative changes in protein biosynthesis in cancer [47]. eIF3A is one major factor controlling eukaryotic translation initiation as part of the eIF3 core complex. Its occurrence is particularly altered in cancers of the lung [48], urinary bladder [19], liver [49] or ovary [50]. Its abundance or genomic alteration has also been correlated with response to chemotherapy [51,52,53]. Therefore, several authors now consider eIF3A to be a proto-oncogene [47].

The amino acid sequence of eIF3A contains a proteasome/COP9 complex/initiation factor (PCI) domain and a spectrin repeat [54] facilitating protein interactions. To date, several interacting proteins have been described [37], including the members of the eIF3 complex, as well as DExH-Box Helicase 29 (DHX29) and others. The recently described detailed structure of the 48S preinitiation complex not only included eIF3 subunits, but also eIF1, eIF1A, eIF4A and eIF4G and RACK1 [9]. As eIF3A has also been reported to control the extracellular signal-regulated kinase [55], a key pathway for proliferation in many tumors, we focused our research on the determination of eIF3A proximal proteins in two fast growing cell lines under normal conditions and under down-regulated protein biosynthesis caused by ER-stress. We were particularly interested in the question of whether significant differences between the cell lines would become obvious, and whether these data might be used for identifying functional differences. HEK-293T cells are adenovirus 5-transformed embryonic kidney cells [56] also carrying a thermosensitive SV40 large T-antigen variant together with neomycin phosphotransferase [57]. These cells are easily transfectable with plasmids, and they are therefore frequently used for recombinant protein expression [58]. The other cell line, the colon carcinoma line HCT-116, has been reported to contain an amplified eIF3A gene, which is why it exhibits an increased abundance of eIF3A mRNA [59].

We decided to perform this investigation using the BioID technique, which takes advantage of a mutated, unspecific biotin transferase fused to a bait protein. As a result, proximal proteins are biotinylated in vivo and can be affinity-purified using the highly specific biotin-avidin interaction, which allows harsh washing procedures. Thus, the BioID technique does not rely on a direct and strong physical interaction of proteins; it reflects the distance of proteins being within a range of about 10 nm, which corresponds approximately to the diameter of a 60 kDa globular protein. It is, therefore, particularly suited for the detection of weak and transient interactions. Nevertheless, this technique relies on a large addition to the bait protein that might influence its biological function. Although we could show that the myc-BirA*-eIF3A fusion has a similar effect on polio-IRES as the unmodified protein, this does not unequivocally exclude the possibility that the fusion protein is impaired in functions unrelated to protein biosynthesis. The alternative construct, the addition of BirA* to the other end of eIF3A, clearly rendered the fusion-protein dysfunctional as there was no specific biotinylation. Other techniques, such as proximity ligation, function within 40 nm [60]; double label immunofluorescence has a spatial resolution of about 200 nm, and resonance energy transfer indicates interactions within 5–10 nm [61]. We therefore regard the BioID technique as a complementary and unbiased method that provides an alternative to detailed structure analysis or co-immunoprecipitation assays. Other studies on the eIF3A interactome used different techniques. In 2009, Sha et al. studied the eIF3A interactome in fission yeast by immunoprecipitation, followed by MS-based peptide identification [10]. They identified about 230 proteins containing many eIFs and a huge set of ribosomal proteins and chaperones. The authors therefore suggested that eIF3A is part of a supercomplex-linking translation to protein degradation.

Using the BioID technique, we detected 84 proteins that were specifically biotinylated in cells expressing the eIF3A-BirA* fusion protein compared to cells expressing the BirA* protein alone. Other studies have previously identified several of these proteins as eIF3A interactors. This applies especially to the eIF subunits 3C, 3D, 3G and 4A1 and the scaffolding protein RACK1, which are included in the detailed structure of the 48S preinitiation complex [9]. In addition, data on the ribosomal proteins and chaperones are in line with the co-immunoprecipitation data from fission yeast [10]. This suggests that the other interacting proteins identified by BioID were indeed also in close proximity to eIF3A in the cell lines tested. The additional immunoprecipitation experiments presented here, confirmed a physical interaction of eIF3A with eIF3D, eIF4A1 and YB-1 but not for PPIA. As stated above, BioID experiments do not rely on a stable and lasting physical interaction of the biotinylated proteins with the bait. Nevertheless, as PPIA is a peptidyl-prolyl-isomerase that only shortly interacts with its substrates, we still suggest a close relationship with eIF3A although IP provided no evidence of a physical interaction. IP experiments are always dependent on the stability of complexes under the selected experimental conditions, which have to lyse the cells securely but are supposed to keep protein interactions intact. The chosen RIPA buffer contains several detergents that might destabilize the protein interaction of eIF3A with PPIA. Immunofluorescence experiments showed that all tested proteins were indeed present in the same compartments as eIF3A namely cytosol and nucleus, suggesting that an interaction is possible.

This list was then further analyzed by computational methods, including over-representation analysis and network analysis. As expected, over-representation analysis revealed that most proteins with a high eIF3A-dependent biotinylation ratio were indeed connected with protein biosynthesis. This holds especially true for members of the eIF3 core proteins. Here, eIF3C, -D and -G were identified. Only one additional initiation factor, eIF4A1, is part of the list. Other constituents of the ribosome were also present, namely constituents of the 60S ribosome (RL23A, RL35, RL37A, RL38, RL39 and RL9). Another associated function is RNA or rRNA binding (14-3-3Z, HNRNPA1, YBX3, HNRNPM, DHX9, LUC7L2, DDX21). Unfolded protein binding involves heat shock proteins HSP-90, HSP-7C, HSPB1, PPIA, NPM1, CCT6A). Surprisingly, there are significantly enriched but unexpected GO-terms, such as cell-cell adhesion or focal adhesion. eIF3A itself has not yet been annotated to these terms. To date, the literature has provided one hint according to which eIF3A might participate in focal adhesions [62]. In that study, Scholler and Kanner cloned p165 using monoclonal antibodies against src-kinase substrates, such as focal adhesion kinase. p165 is identical to eIF3A; however, we are not aware of any further links to focal adhesions. Altogether, according to the QuickGO database (https://www.ebi.ac.uk/QuickGO/), about 839 human proteins are annotated to focal adhesion, including several ribosomal proteins. The evidence for this annotation is given by “high-throughput direct assay evidence used in manual assertion”, which we regard as not being very reliable. In conclusion, we think that eIF3A is not a part of a focal adhesion.

There are also differences between the two cell lines. First, HEK-293T cells showed higher numbers of interacting proteins than the HCT-116 cells, although the auto-biotinylation of eIF3A was comparable. This might reflect differences in cellular physiology between the fast-growing embryonic cell line and the colon-derived cancer cell line. The bicistronic luciferase vector, used for testing differences in cap to Polio-IRES-mediated translation mode, detected only a small but statistically significant difference between the cell lines. However, other IRES sequences could be more strongly affected. More studies with cancer cells are necessary to obtain reliable evidence for such a hypothesis, especially concerning a higher impact of IRES-mediated translation initiation. There are only a few proteins specifically labeled in one cell line. We are particularly interested in proteins appearing only in the colon cancer cell line. These were PPIA, PROF1, PTMA and RS30. PPIA, which is important for proper protein folding, was found to be dysregulated in colon cancer [63] and to influence signaling [64]. However, Western blot detected no obvious differences in protein abundance, while immunofluorescence showed that PPIA is more granular in HEK-293T than in HCT-116 cells (Appendix A); this difference in localization could be a sign of functional differences. A close association with eIF3A could indicate a coupling to protein folding, in particular proline-isomerization and translation in the colon carcinoma cell line. It is known that protein folding starts already at the ribosome [65,66], and the loss of eIF3A was reported to affect the folding of the cystic fibrosis protein CFTR [67]. PTMA is a histone- and transcription factor-binding protein that regulates IP-3 formation and androgen receptor signaling. It might therefore be important for the cancer cell physiology [68], but a relation to protein biosynthesis has not yet been described. In contrast to PPIA, nucleophosmin (NPM1) showed a higher biotinylation ratio in HEK-293T cells than in the HCT-116 cancer cell line. NPM1 has already been identified as an interacting protein of the eIF2- kinase PKR [69] and is involved in ribosome biogenesis and maintenance of the nucleolus [70]. Again, this suggests a close spatial relationship to eIF3A.

As we presumed that BioID identifies false positive interacting proteins, we decided to analyze the effect of thapsigargin on the interactome as control treatment. Thapsigargin induces ER stress by blocking the SERCA Ca^2+^ transporters. These transporters translocate cytosolic calcium ions back into the ER using ATP hydrolysis as energy source. Blocking of SERCA results in a rapid loss of Ca^2+^ from the ER to the cytosol, which causes the accumulation of misfolded proteins in the ER. The resulting unfolded protein response (UPR) is characterized by high expression of chaperones such as GRP78. UPR also results in a shutdown of protein biosynthesis mediated by activation of inositol requiring protein-1 alpha (IRE1-alpha), activating transcription factor 6 (ATF6), and protein kinase RNA (PRK)-like ER-associated kinase (PERK). Increased phosphorylation of eIF2α phosphorylation by eIF2α-kinases such as PERK inhibits protein synthesis. Phosphorylation of eIF2α blocks the formation of the ternary complex, consisting of methionyl-tRNA, eIF2, and GTP. This is an early step in translation initiation. As a result, the 48S preinitiation complex cannot be formed.

We therefore expected that under thapsigargin-induced ER-stress, eIF3A-proximal proteins involved in translation are lost, and might then be replaced by other proteins, indicating an altered function of eIF3A. However, only a few proteins still showed a significant biotinylation under ER-stress (RS17, RS29, RBM39) or biotinylation appeared only under stress (DCD, H4), but in only one cell line at one time point investigated. As thapsigargin depletes calcium-ions from the ER to the cytosol, this substance also increases cytosolic Ca^2+^ concentration. This effect was quantified by Huang and coworkers in a prostate cancer cell line, which rose from 50 to 100 nM [71]. In vitro reconstitution experiments have shown that eIF3 complexes associate without additional Ca^2+^ or added calcium chelators (see, e.g., [72]). Nevertheless, we cannot exclude that other eIF3A protein interactions are dependent on calcium-ions or calcium-ion mediated signaling events.

Some other identified proteins are not expected to be eIF3A interactors due to their predominant cellular location. ATPO and CH60 should be solely located in the mitochondrial inner membrane or matrix. If this is correct, an interaction is possible only when the mitochondrial membrane is damaged, i.e., in the course of apoptosis. This might be possible as thapsigargin induces apoptosis; however, CH60 (HSPD1) was labeled in untreated cells only. Interestingly, the compartments database [73] suggests that this protein is also present in cytosol and the nucleus (https://compartments.jensenlab.org/), and similar expectations apply to ATPO. Our data therefore suggest a potential function of eIF3A in relation to these proteins. Nevertheless, more detailed experiments are required to thoroughly understand this result.

The BioID method is not free of drawbacks that are important for our study. It has been reported that naturally occurring biotinylated proteins are not only purified by the streptavidin affinity method, but tend to be significantly enriched in BioID experiments. There is also a problem with keratins, which are often contaminants in protein preparations due to their omnipresence. As a consequence, such proteins are usually not considered to be true interactors.

In addition, ribosomal proteins have also been described to be enriched in BioID experiments [39]. In the case of eIF3A-birA* transfections, however, ribosomal proteins are expected as positive hits and were indeed included in our list of presumptively interacting proteins that were subjected to a strict statistical evaluation. Strikingly, according to our data, ribosomal proteins, in particular, are still over-represented under the influence of thapsigargin. We can therefore not exclude that in our study, these proteins represent an artefact, which means that eIF3A has even fewer interacting proteins when protein biosynthesis is inhibited by ER-stress.

## 4. Materials and Methods

### 4.1. Plasmids and Cloning

The plasmids pcDNA3.1-mycBioID and pcDNA3.1 MCS-BirA(R118G)-HA containing the mutant BirA-R118G (BirA*) were developed by Roux et al., 2012 [39,74] and were purchased from Addgene (Plasmid #35700 and 36047). The eIF3A coding region was amplified by PCR from a full-length clone (accession number BC114429, BioCat, Heidelberg, Germany) using the following primers containing BamHI restriction sites and overlapping vector sequences to facilitate “cold-fusion-cloning” (Biocat, Heidelberg, Germany). Forward: CCACACTGGACTAGTGGATCCATGccggcctattttcagag, reverse: CTTGGTACCGAGCTCGGATCCttaacgtcgtactgtggtcc for c-myc BirA* fusion and ctccggattcgaattcggatccatgccggcctattttcagag (forward) and cacggtgttgtccttggatccacgtcgtactgtggtccatc (reverse) for the BirA*-HA fusion. For cloning, both plasmids were linearized with BamHI (Promega) and incubated with the cold fusion cloning reagents according to the manufacturer’s recommendations. Clones were selected based on restriction pattern and subjected to DNA sequencing (ABI prism) to check for possible mutations caused by possible infidelity of the PCR enzyme (Accuprime polymerase, Thermo-Fisher) used. Expression of target sequences in this plasmid is under the control of the cytomegalovirus (CMV) immediate-early enhancer/promoter. These vectors were used for transient expression in HCT-116 and HEK-293T cells. The vectors pcDNA3.1-mycBioID and pcDNA3.1 MCS-BirA(R118G)-HA containing only myc-BirA* or HA-tagged BirA* were used for control transfections. Only the myc-fusion construct showed significant, specific biotinylation in BioID experiments, the HA-BirA* eIF3A fusion vector failed for this purpose. For determination of the ratio of cap- to Polio-IRES dependent translation a bicisstronic vector (pcDNA3-RLUC-POLIRES-FLUC, Addgene) was used [75]. Renilla luciferase is translated in a cap-dependent manner, whereas firefly luciferase depends on the use of the IRES sequence.

### 4.2. Cell Lines and Cell Culture

HCT-116 and HEK-293T cells were routinely maintained in RPMI 1640, which contained 10% fetal bovine serum (FBS) and 1% Penicillin-Streptomycin (all from Biochrom, Berlin, Germany). Cells were grown at 37 °C in an atmosphere of 95% air and 5% CO_2_ and transferred weekly into new flasks (Nunc) after detachment with Trypsin/EDTA (Biochrom). For BioID experiments, cells were seeded to a density of 30% in 6-well plates, whereas for luciferase reporter gene assays, 10,000 cells per well were seeded into white 96-well plates with transparent bottoms (µClear, Greiner-Bio-One). For luciferase detection, the cells were lysed the day after transfection with 20 µL passive lysis buffer (Promega) and a dual-luciferase assay was performed as described earlier [76,77] in a BMG Clariostar microplate reader. Ratio of the luciferase activities were determined and normalized to one cell line.

### 4.3. Proximity-Dependent Protein Labeling Methods—Sample Preparation for Mass Spectrometry

myc-BirA*-eIF3A and myc-BirA* vectors were transiently transfected into the cells using Lipofectamin LTX with PLUS^TM^ reagent (Thermo Scientific, San Jose, CA, USA) in Biotin free MEM with Earle’s salts medium (Merck Millipore, Darmstadt, Germany) for 24 h at 37 °C in an atmosphere of 95% air and 5% CO_2_. BirA* control and BirA*-eIF3A fusions transfected cells were treated with 50 µM Biotin (Serva, Heidelberg, Germany) for 6 h or 24 h to induce biotinylation of proteins. Thapsigargin treatment was done at 1 µM concentration for 3 h before addition of biotin for 6 h or 24 h. Before harvesting, cells were washed twice in MEM with Earles’s salts medium for biotin removal. Cells were suspended in a RIPA lysis buffer (50 mM Tris/HCl pH 7.5, 5 mM EDTA, 100 mM NaCl, 10 mM K_2_HPO_4_, 10% Glycerol, 1% Triton X-100, 0.5% NP-40, containing phosphatase and protease inhibitor cocktails (Sigma Aldrich) for 30 min on ice. Cellular debris was removed by centrifugation for 10 min at 15,000 g and 4 °C. Protein lysate was mixed with Streptavidin magnetic beads (Thermo Scientific) and the binding was processed for 1h at room temperature. After binding, the beads were washed step by step with the following solvents: 8 M Urea, 1% Triton, 0.5 M NaCl, 20% Acetonitrile, 50 mM NH_4_HCO_3_. Then, the beads with the biotinylated proteins were stored in 50 mM NH_4_HCO_3_. A portion of the sample was run on denaturing SDS-PAGE, transferred to nitrocellulose membrane and probed with anti-Biotin antibody to confirm the biotinylation.

### 4.4. Mass Spectrometry

Sample preparation for mass spectrometry was performed by on-beads digestion for best analytical sensitivity. The procedure in brief: Beads were rehydrated in 50 mM NH_4_HCO_3_, pH 8.0 and subsequently incubated with 1 mM DTT at 56 °C for 45 min. Afterwards, reduced cysteins were β-methylthiolated by addition of 5 mM MMTS (methyl-methane-thiosulfonate) at room temperature for 30 min. Proteins were digested by addition of 0.5 µg trypsin (Promega, Trypsin Gold) and incubation at 37 °C overnight. The resulting tryptic peptides were gathered by collection of the supernatants combined with two washing steps of the beads using 50 µL of 25 mM NH_4_HCO_3_ each. All supernatants of a sample were pooled and dried down in a vacuum centrifuge. The peptides were redissolved in 5 µL 0.1% trifluoroacetic acid (TFA) and purified on ZIP-TIP, C18-nanocolumns (Millipore, Billerica, MA, USA). Peptides were eluted in 7 µL 70% (*v*/*v*) acetonitrile (ACN) and subsequently dried in a vacuum centrifuge.

LC-MS/MS was performed on a hybrid dual-pressure linear ion trap/orbitrap mass spectrometer (LTQ Orbitrap Velos Pro, Thermo Scientific, San Jose, CA, USA) equipped with an EASY-nLC Ultra HPLC (Thermo Scientific, San Jose, CA, USA). Peptide samples were dissolved in 10 µL 2% ACN/0.1% trifluoric acid (TFA) and fractionated on a 75 μm I.D., 25 cm Fortis C18-column, packed with 1.7 µm resin (Fortis Technologies LtD., GB). Separation was achieved by applying a gradient from 2% ACN to 35% ACN in 0.1% formic acid over a 150 min gradient at a flow rate of 300 nL/min.

The LTQ Orbitrap Velos Pro MS has exclusively used CID-fragmentation when acquiring MS/MS spectra consisted of an orbitrap full MS scan followed by up to 15 LTQ MS/MS experiments (TOP15) on the most abundant ions detected in the full MS scan. Essential MS settings were as follows: full MS (FTMS; resolution 60.000; m/z range 400–2000); MS/MS (Linear Trap; minimum signal threshold 500; isolation width 2 Da; dynamic exclusion time setting 30 s; singly charged ions were excluded from selection). Normalized collision energy was set to 35%, and activation time to 10 ms. Relative abundance of compounds was determined using the Progenesis QI-software (Nonlinear Dynamics, Newcastle upon Tyne, UK).

### 4.5. Immunoprecipitation and Immunoblotting

A whole cell extract was prepared and analyzed by Western blotting essentially as described previously [41]. The following additional antibodies were used: eIF3A (Cat# 3411), Cell Signaling (Danvers, MA, USA); β-actin (Cat# A5441, Sigma-Aldrich (Steinheim, Germany). Detection of the chemiluminescent signal was done in an INTAS chemstar imager (Intas, Göttingen, Germany) using an enhanced chemiluminescent substrate ((MERCK Millipore, Darmstadt, Germany).

For immunoprecipitation, cells were transfected with the plasmids as described for BioID-experiments. Cell lysates of transfected cells were prepared with RIPA buffer (50 mM Tris-HCl, pH 7.5, 100 mM NaCl, 5 mM EDTA, 1% Triton X-100, 10% glycerol, 10 mM K_2_HPO_4_, 0.5% NP-40, 1× protease inhibitor mixture (Roche Diagnostics GmbH, Mannheim, Germany), 1 mM Na_3_VO_4_, 1 mM Na_2_MoO_4_, 20 mM NaF, 0.1 mM PMSF, 20 mM β-glycerol-2-phosphate). Cell lysates were incubated with Pierce ^TM^ Anti-c-Myc magnetic beads (Thermo Scientific, Rockford, IL, USA) overnight at 4 °C to capture the immunocomplexes. Beads were washed three times with RIPA buffer before subjected to SDS-PAGE. Samples were boiled in sample buffer (50 mM Tris/Cl, pH 6.8, 2% SDS, 10% glycerol, 100 mM dithiothreitol, and 0.1% bromophenol blue), subjected to SDS-PAGE (NuPAGE 4–12%, MOPS-SDS or MES-SDS running buffers were used to achieve different separation ranges, Invitrogen) and electrotransferred to Immobilon-P transfer polyvinylidene fluoride membranes (Millipore, Schwalbach, Germany) by tank blotting. The blots were probed with the following primary antibodies diluted in 5% BSA in TBS/Tween 20 (c-myc: Invitrogen (mouse #132500) (rabbit #700648), eIF3D: Genetex (GTX101424), eIF4A1 CellSignaling (#2490), YB-1: Eurogentec (EP052153 and EP081818), Cyclophilin A: abcam (ab58144)). HRP-conjugated secondary antibodies and Pierce ECL Plus (Thermo Scientific, Rockford, IL, USA) substrates were used for antigen detection using X-ray film (AGFA Curix HT1.000G PLUS, Agfa Healthcare, Mortsel, Belgium).

### 4.6. Immunofluorescence

Cells were seeded on chamber glass slides (Sarstedt, Nuembrecht, Germany or Greiner BioOne, Frickenhausen, Germany) to approximately 50% confluence. After overnight incubation in full medium, cells were treated with solvent or thapsigargin for 6 h and then washed with PBS and fixed with formaldehyde (4% in PBS) for 1 h at 4 °C. Afterwards, the slides were blocked with 5% normal goat serum in PBS before incubation with the primary antibodies directed against eIF3A (#3411, Cell Signaling (Danvers, MA, USA) or St. Cruz Biotech (sc-365789), nucleophosmin (Abcam, ab10530), Cyclophilin (Abcam, ab58144), fluorescein-streptavidin (Vector Laboratories, via Biozol, Germany), eIF4A1 (Cell Signaling, 2490) YB-1 (Eurogentec EP052153 and EP081818), and eIF3D (Genetex, GTX101424) at 4 °C overnight in a humidified chamber. After washing three times with PBS/Tween20 (0.05%), secondary antibodies (DyLight 488 and -549, Thermofisher) were added for 1h at RT and after three further washes, the slides were covered in DAPI containing mounting medium.

Samples were analyzed using an inverted Confocal Microscope System Leica SP8 (Leica Mannheim, Germany) equipped with a Plan Apo 63x/1.4 oil objective and controlled by LAS X software (Leica). For immunofluorescence using the St. Cruz Biotech antibody sc-365789, a Plan Apo 63x/1.2 water objective was applied. To avoid bleed through between the different color channels, sequential unidirectional scanning was performed at 400 Hz using the following settings. Sequence 1: excitation 488 nm, Emission 495 nm–532 nm combined with transmitted light detection, sequence 2: excitation 405 nm, emission 411 nm–464 nm and excitation 561 nm, emission 566 nm–620 nm. The sequences were altered between lines. Voxel size was adjusted to 82 nm × 82 nm × 299 nm (dx, dy, dz) to fit to Nyquist theorem. Images of the channels were pseudo colored; DAPI (excitation 405 nm) in blue, DyLight488 (excitation 488 nm) in green and DyLight546 (excitation 561 nm) in red. Single planes out of the data stacks were visualized using ImageJ software [78].

Standard fluorescence microscopy was done using a Zeiss Axioplan 2 microscope, with a Plan Neofluar 40×/0.75 objective (Zeiss, Jena, Germany) and Zeiss filter sets 02 (G365 nm, FT 395 nm, LP > 420 nm) −10 (BP 450–490 nm, FT 510 nm, BP 515–565 nm), and −15 (BP 546 nm, FT 580 nm, LP >590 nm). Microscopic images were taken with a JAI CV-M1 digital camera as part of the ISIS imaging system version 4.4.24 (Meta-Systems, Altlussheim, Germany) [79].

### 4.7. Overrepresentation Analysis

Overrepresentation analysis of protein lists for gene ontology terms was done using the web-based Genecodis platform [42,80,81]. At least three protein hits were required for a GO-term to be included in this study.

### 4.8. Network Visualization using Cytoscape

Cytoscape vers. 3.8.0. [43] with StringApp and Legendcreator were used to visualize, expand and analyze the interaction network. The interaction table for HCT-116 and HEK-293T cells was formatted with eIF3A as source node and the interaction partners as target nodes. We then used the String database for network enrichment with a selectivity of 0.5. Cytoscape was also used for over-representation analysis for Gene ontology (GO) terms, the KEGG pathway database and Reactome and visualized as pie chart around the nodes (Figure 5).

### 4.9. Statistical Analysis

For mass spectrometry results, raw data processing, protein identification and PTM assignment of the high resolution orbitrap data sets was performed by the de novo- sequencing algorithms of PEAKS Studio 8.0 (Bioinformatics Solutions). False discovery rate (FDR) was set to < 1%. Data are reported for at least three independent experiments unless otherwise indicated. The statistical significance of the difference between the determinations was calculated by analysis of variance using ANOVA, Tukey–Kramer multiple comparisons test or Student’s t-test. The difference was considered as significant when the *p* value was < 0.05.

## 5. Conclusions

By using the BioID technique we identified 84 proteins that could be specifically labeled by biotin and are therefore suspected to be eIF3A interacting proteins. The majority of these proteins are known interactors, but several have not been associated with eIF3A before. This result suggests that eIF3A is involved in previously undescribed functions i.e., protein folding, which should be the focus of future research.

## Figures and Tables

**Figure 1 cancers-13-01293-f001:**
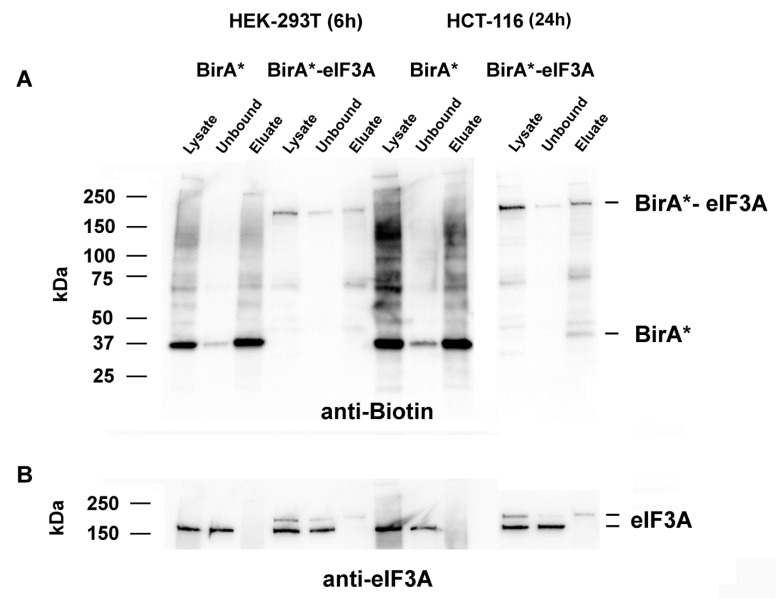
Biotinylation of proteins by myc-BirA* and myc-BirA*-eIF3A in HEK-293T and HCT-116 cells. Cells were transfected with the plasmids and incubated with biotin for the indicated time. Crude lysate and streptavidin-affinity purified proteins were analyzed by Western blot, demonstrating the presence of biotinylated proteins (**A**) and eIF3A (**B**) as well as the myc-BirA*-eIF3A fusion protein.

**Figure 2 cancers-13-01293-f002:**
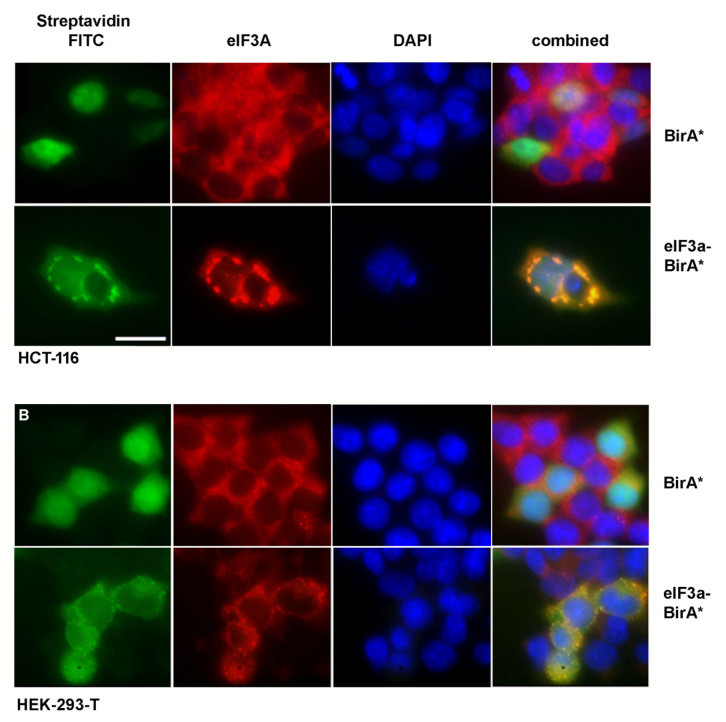
Visualization of biotinylated proteins and eIF3A by immunofluorescence. HCT-116 (**A**) and HEK-293T (**B**) cells were seeded on glass slides, fixed and stained for biotinylated proteins and eIF3A by streptavidine-fluorescein conjugate or indirect immunofluorescence, respectively, as described in the “Materials and Methods” section. Nuclei were counterstained by DAPI. Scale bar indicates 50 µm.

**Figure 3 cancers-13-01293-f003:**
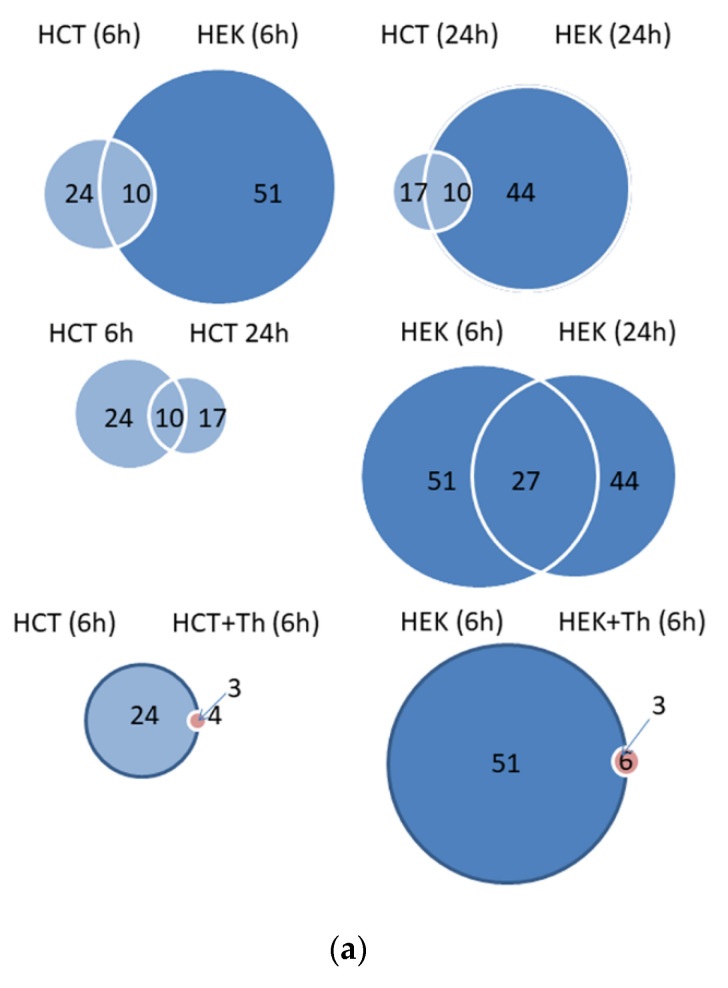
Results of the mass-spectrometric identification of biotinylated proteins in HEK-293T and HCT-116 after 6 and 24 h of labeling with biotin. MS data were analyzed as described in the method section. Only proteins that were significantly enriched at least 1.5-fold in the myc-BirA*-eIF3A transfections, in comparison to BirA* transfection for at least one experimental condition, are included. (**a**) Diagrams comparing identified proteins between the experimental conditions. (**b**) Heat map illustrating the eIF3A-dependent biotinylation ratio for all identified interacting proteins under all experimental conditions.

**Figure 4 cancers-13-01293-f004:**
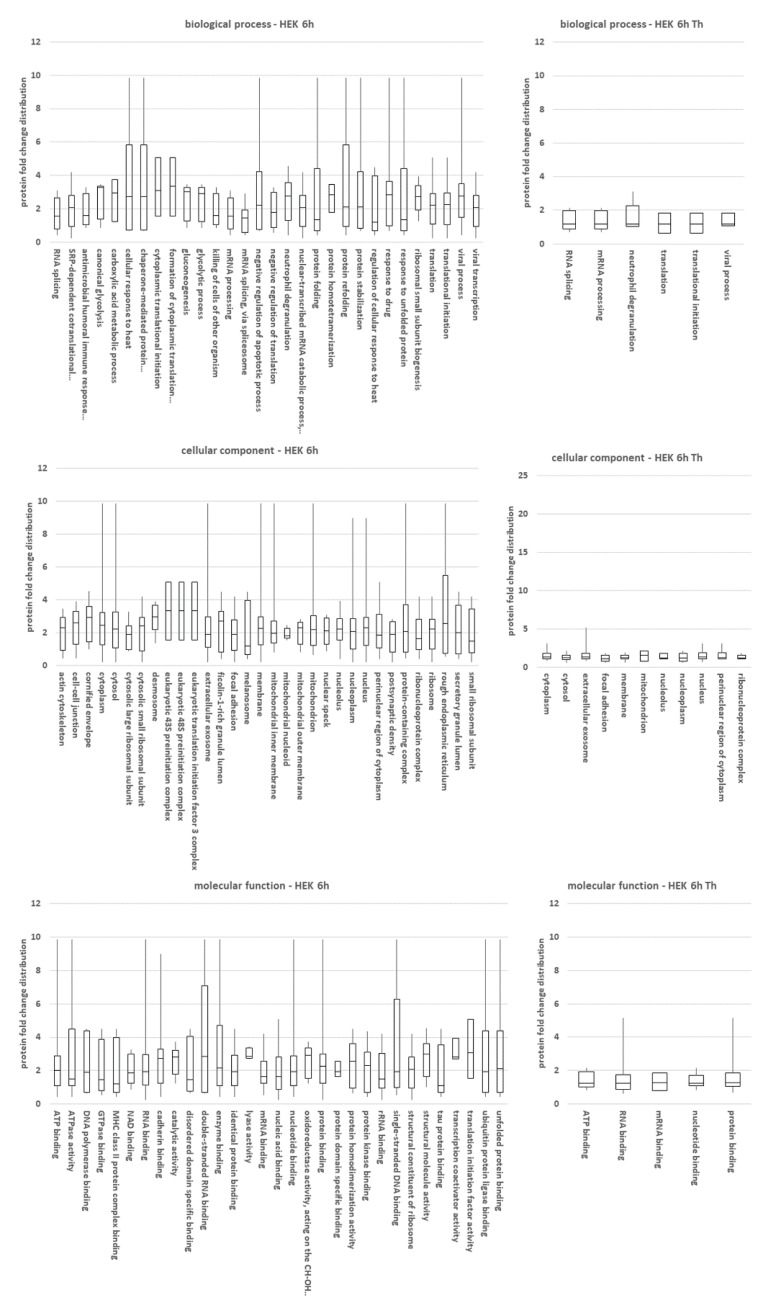
Overrepresentation analysis of protein hits using genecodis. The eIF3A-dependent biotinylation ratios for the proteins are presented as box plots for HEK-293T cells at the 6 h labeling time point. Data for eIF3A were omitted as the high score for this protein biased box plots containing eIF3A. Box plots for HCT-116 and other time points can be found in the Appendix A.

**Table 1 cancers-13-01293-t001:** Gene ontology term over-representation analysis. eIF3A-interacting proteins were analyzed on the genecodis webpage for overrepresentation of GO-terms in biological processes, cellular components and molecular functions. Only hits with an adjusted *p*-value less than 0.01 are shown.

**Term**	**Annotation**	**Genes Found**	**Genes in Term**	**Hyp._pval_adj.**
**Biological Process**				
translational initiation	GO:0006413	18	147	5.52e × 10^−19^
translation	GO:0006412	21	315	2.94e × 10^−17^
SRP-dependent cotranslational protein targeting to membrane	GO:0006614	13	89	1.24e × 10^−14^
viral transcription	GO:0019083	13	109	1.44e × 10^−13^
nuclear-transcribed mRNA catabolic process, nonsense-mediated decay	GO:0000184	13	117	2.98e × 10^−13^
cytoplasmic translational initiation	GO:0002183	5	17	8.87e × 10^−7^
viral process	GO:0016032	15	577	1.59e × 10−^6^
formation of cytoplasmic translation initiation complex	GO:0001732	4	16	4.93e × 10^−5^
neutrophil degranulation	GO:0043312	12	484	5.46e × 10^−5^
viral translational termination-reinitiation	GO:0075525	3	5	5.56e × 10^−5^
response to unfolded protein	GO:0006986	6	74	5.76e × 10^−5^
protein stabilization	GO:0050821	8	191	7.58e × 10^−5^
protein refolding	GO:0042026	4	20	7.98e × 10^−5^
gluconeogenesis	GO:0006094	5	45	7.99e × 10^−5^
negative regulation of apoptotic process	GO:0043066	12	547	1.43e × 10^−4^
glycolytic process	GO:0006096	4	36	7.05e × 10^−4^
cellular response to interleukin-7	GO:0098761	3	12	7.49e × 10^−4^
cytoplasmic translation	GO:0002181	4	42	1.24e × 10^−3^
carboxylic acid metabolic process	GO:0019752	3	18	2.30e × 10^−3^
protein folding	GO:0006457	6	175	3.33e × 10^−3^
ribosomal small subunit biogenesis	GO:0042274	3	21	3.36e × 10^−3^
chaperone-mediated protein complex assembly	GO:0051131	3	21	3.36e × 10^−3^
canonical glycolysis	GO:0061621	3	27	6.09e × 10^−3^
chaperone-mediated protein folding	GO:0061077	3	30	6.98e × 10^−3^
establishment of cell polarity	GO:0030010	3	35	9.78e × 10^−3^
**Cellular Component**				
**Term**	**Annotation**	**Term_Genes_Found**	**Term_Genes**	**hyp_pval_adj**
extracellular exosome	GO:0070062	45	2170	1.70e × 10^−20^
cytosol	GO:0005829	59	5175	1.33e × 10^−16^
focal adhesion	GO:0005925	20	418	2.06e × 10^−14^
ribosome	GO:0005840	15	237	2.16e × 10^−12^
cytosolic small ribosomal subunit	GO:0022627	8	47	7.04e × 10^−10^
ribonucleoprotein complex	GO:1990904	11	160	1.93e × 10^−9^
nucleus	GO:0005634	55	6594	4.74e × 10^−9^
cytoplasm	GO:0005737	52	6873	1.18e × 10^−6^
ficolin-1-rich granule lumen	GO:1904813	8	124	1.20e × 10^−6^
cytosolic large ribosomal subunit	GO:0022625	6	59	3.43e × 10^−6^
eukaryotic 48S preinitiation complex	GO:0033290	4	15	7.32e × 10^−6^
eukaryotic translation initiation factor 3 complex	GO:0005852	4	16	8.92e × 10^−6^
eukaryotic 43S preinitiation complex	GO:0016282	4	17	1.07e × 10^−5^
mitochondrial nucleoid	GO:0042645	5	44	1.62e × 10^−5^
cornified envelope	GO:0001533	5	45	1.70e × 10^−5^
small ribosomal subunit	GO:0015935	4	29	8.39e × 10^−5^
secretory granule lumen	GO:0034774	6	115	1.05e × 10^−4^
nucleoplasm	GO:0005654	32	3719	1.27e × 10^−4^
polysomal ribosome	GO:0042788	4	34	1.29e × 10^−4^
mitochondrion	GO:0005739	19	1553	1.30e × 10^−4^
vesicle	GO:0031982	7	183	1.30e × 10^−4^
melanosome	GO:0042470	5	99	5.70e × 10^−4^
membrane	GO:0016020	49	7884	1.11e × 10^−3^
desmosome	GO:0030057	3	26	1.51e × 10^−3^
protein-containing complex	GO:0032991	11	745	1.94e × 10^−3^
perinuclear region of cytoplasm	GO:0048471	10	722	5.48e × 10^−3^
postsynaptic density	GO:0014069	6	282	7.45e × 10^−3^
nuclear matrix	GO:0016363	4	109	7.67e × 10^−3^
cell-cell junction	GO:0005911	5	200	9.23e × 10^−3^
**Molecular function**				
**Term**	**Annotation**	**Term_Genes_Found**	**Term_Genes**	**hyp_pval_adj**
RNA binding	GO:0003723	47	1531	4.51e × 10^−28^
cadherin binding	GO:0045296	16	316	5.24e × 10^−11^
structural constituent of ribosome	GO:0003735	13	203	3.62e × 10^−10^
unfolded protein binding	GO:0051082	8	115	2.51e × 10^−6^
mRNA binding	GO:0003729	9	217	2.31e × 10^−5^
rRNA binding	GO:0019843	5	45	6.81e × 10^−5^
protein folding chaperone	GO:0044183	4	27	2.16e × 10^−4^
translation initiation factor activity	GO:0003743	5	60	2.17e × 10^−4^
protein binding	GO:0005515	71	12556	4.09e × 10^−4^
double-stranded RNA binding	GO:0003725	5	72	4.27e × 10^−4^
NAD binding	GO:0051287	4	39	6.20e × 10^−4^
oxidoreductase activity, acting on the CH-OH group of donors, NAD or NADP as acceptor	GO:0016616	4	43	8.42e × 10^−4^
MHC class II protein complex binding	GO:0023026	3	16	9.24e × 10^−4^
nucleotide binding	GO:0000166	20	1781	1.07e × 10^−3^
ATPase activity	GO:0016887	7	237	1.43e × 10^−3^
DNA polymerase binding	GO:0070182	3	20	1.51e × 10^−3^
ribosome binding	GO:0043022	4	55	1.58e × 10^−3^
single-stranded DNA binding	GO:0003697	5	111	1.80e × 10^−3^
mRNA 5’-UTR binding	GO:0048027	3	26	2.16e × 10^−3^
ATP binding	GO:0005524	17	1489	2.42e × 10^−3^
protein homodimerization activity	GO:0042803	11	686	2.48e × 10^−3^
ubiquitin protein ligase binding	GO:0031625	7	298	3.25e × 10^−3^
catalytic activity	GO:0003824	9	506	3.40e × 10^−3^
transcription coactivator activity	GO:0003713	7	306	3.55e × 10^−3^
miRNA binding	GO:0035198	3	34	3.91e × 10^−3^
GTPase binding	GO:0051020	3	35	4.13e × 10^−3^
disordered domain specific binding	GO:0097718	3	38	4.97e × 10^−3^
lyase activity	GO:0016829	5	167	6.45e × 10^−3^
tau protein binding	GO:0048156	3	45	7.51e × 10^−3^
chaperone binding	GO:0051087	4	105	8.40e × 10^−3^

## Data Availability

Data is contained within the article or Appendix A.

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
