# Peer review of "Interactome Mapping of eIF3A in a Colon Cancer and an Immortalized Embryonic Cell Line Using Proximity-Dependent Biotin Identification"

_cancers, 2021, doi:10.3390/cancers13061293_

Round 1
Reviewer 1 Report
Since the last submission authors have strong progress in enhancing the quality of the manuscript, such as by adding a Western blot in Figure S2 that meets the requirements for publication. Also, I appreciate the attempt to check the colocalization of eIF3a with YB-1, eIF3D, eIF4A1 by immunofluorescence. However, there is one remaining flaw that prevents a recommendation for publication in the MDPI Cancers journal.
- In Supplementary Figures 5A and 5B eIF3a show almost exclusively cytoplasmic localization, while in Figures 5C, D nuclear. As a result, the reader observes an inverse correlation between eIF3a and YB-1, eIF3D, & eIF4A1. The authors explain the different localization of eIF3a by using antibodies from different suppliers. However, these results also show that one of the antibodies is potentially not specific.
I would strongly recommend checking the specificity of the eIF3a antibody by any available technique (Western blots +/- knockdown or knockout of eIF3a), and then repeat the co-localization experiment (at least one or several of Supplementary Figures 5A, B, C, or D) with an antibody of proven specificity.
Reviewer 2 Report
The authors have provided the requested data for PPIA in Supplemental Figure 3. One minor suggested change is to indicate in the Results section: "... we failed to obtain a specific signal in the co-immunoprecipitation for PPIA," since there is significant signal in both eIF3A and control lanes for the PPIA Western blot.
Author Response
Reviewer 2:
The authors have provided the requested data for PPIA in Supplemental Figure 3. One minor suggested change is to indicate in the Results section: "... we failed to obtain a specific signal in the co-immunoprecipitation for PPIA," since there is significant signal in both eIF3A and control lanes for the PPIA Western blot.
Thank you for this remark. We have changed the sentence accordingly.

This manuscript is a resubmission of an earlier submission. The following is a list of the peer review reports and author responses from that submission.
Round 1
Reviewer 1 Report
The manuscript is much improved. All concerns have been satisfactorily addressed.
Author Response
Reviewer 1
The manuscript is much improved. All concerns have been satisfactorily addressed.
We thank the reviewer for this kind decision.
Reviewer 2 Report
The quality of the revised paper has been improved. This paper is now acceptable for the publication in Cancers
Author Response
Reviewer 2
The quality of the revised paper has been improved. This paper is now acceptable for the publication in Cancers
We thank the reviewer for this kind decision.
Reviewer 3 Report
Since the last submission the authors have done a good job of improving the manuscript by adding several key controls such as Ips of several key proteins. However, there are still several flaws that prevent a recommendation for publication in the MDPI Cancers journal.
- In additional to the biochemical experiments, visualization of the co-localization of the top targets (eIF3D; YB-1; eIF4A1) with eIF3A by immunofluorescence would provide reasonable evidence for the specificity of the BioID technique.
- The quality of the Western blots from Figure S2 do not meet the requirements for a scientific publication. For all phosphoprotein blots, the levels of the total protein must be shown to make any conclusions Also, authors should choose adequate loading control that do not change during the treatments applied.
Author Response
Reviewer 3
Since the last submission the authors have done a good job of improving the manuscript by adding several key controls such as Ips of several key proteins. However, there are still several flaws that prevent a recommendation for publication in the MDPI Cancers journal.
- In additional to the biochemical experiments, visualization of the co-localization of the top targets (eIF3D; YB-1; eIF4A1) with eIF3A by immunofluorescence would provide reasonable evidence for the specificity of the BioID technique.
According to this critic, we have performed additional immuno-fluorescence co-localization experiments. For these, we decided to use a mouse monoclonal for the detection of eIF3A, because we had the other antibodies already available as polyclonal rabbit antisera. However, when changing from a polyclonal serum to a monospecific monoclonal antibody that might recognize a different epitope, it could be expected that result will look different. Such effects can arise from a different distribution or accessibility of the different epitopes in the cellular compartments. Indeed, the monoclonal anti-eIF3A antibody produced more signal in the nucleus than the polyclonal serum. However, eIF3A is expected to be present in both compartments, so we regard this result as conceivable. The putative interacting proteins YB-1, eIF3D and eIF4A1 showed a similar localization, however the signal was mainly, but not exclusively, in the cytosol. Concerning YB-1, it should also be taken into consideration that the acetylated protein is found in the nucleus. Taken together, we think that the additional immuno-fluorescence data support the idea of an interaction with eIF3A.
We have added the images into the supplement and referenced and discussed the results in the manuscript.
- The quality of the Western blots from Figure S2 do not meet the requirements for a scientific publication. For all phosphoprotein blots, the levels of the total protein must be shown to make any conclusions Also, authors should choose adequate loading control that do not change during the treatments applied.
Thank you for this point, we have definitely not taken enough care for this experiment. We have now repeated the experiment twice and determined the unphosphorylated proteins as well and performed a quantification of the bands. For p-PERK, we obtained no specific signal with the antibody used, we therefore show the phosphorylation-dependent band shift as we did in our previous paper (Vo et al. Biomolecules 2019, 9, 503; doi:10.3390/biom9090503). Additionally, we included CHOP as ER-stress marker into the protocol. Interestingly, CHOP was only induced in the HCT-116 cell line, which also exhibited a more significant phosphorylation of IRE1α in comparison to HEK-293T. There was only a slight increase in relative eIF2α phosphorylation in these experiments. We have modified the supplemental figure S1 accordingly and added this information to the manuscript.
Reviewer 4 Report
The authors have now provided co-immunoprecipitation validation for some of the identified interacting partners of eIF3A, as requested. In the manuscript text they indicate that a co-IP for PPIA was also performed, but no signal for PPIA could be detected. For completeness, the authors should include the blot(s) for PPIA in the final figure.
Author Response
Reviewer 4
The authors have now provided co-immunoprecipitation validation for some of the identified interacting partners of eIF3A, as requested. In the manuscript text they indicate that a co-IP for PPIA was also performed, but no signal for PPIA could be detected. For completeness, the authors should include the blot(s) for PPIA in the final figure.
We thank the reviewer for this remark. We have now added the result for PPIA into the supplemental figure S3.